# Interactive Cell Detection in H&E-stained slides of Diffuse Gastric Cancer

**Robin Lomans**                    robin.ooijer@radboudumc.nl
**Rachel van der Post**             chella.vanderpost@radboudumc.nl
**Francesco Ciompi**                francesco.ciompi@radboudumc.nl
*Department of Pathology, Radboudumc, Nijmegen, the Netherlands*

## Abstract

We present an interactive detection model to improve the cell annotation workflow of diffuse gastric cancer. The model relates image and user inputs and is trained to detect three types of cells in diffuse gastric cancer histology. We measure model multi-class cell detection performance as per-class F1 score and we show that it increases with the number of user input clicks. Moreover, we show that the proposed interactive annotation approach substantially reduces the number of required user actions needed for complete image annotation, achieving a 17% reduction for the multi-class case. Future work will implement an iterative approach to filter out recurring false positives for further performance improvement.

**Keywords:** Interactive detection, signet ring cells, diffuse gastric cancer, interactive annotation.

## 1. Introduction

Diffuse gastric cancer (DGC) is a type of cancer characterized by diffusely infiltrating signet ring cell (SRC) carcinomas, which are difficult to detect due to their focality and diffuse infiltration between normal mucosal epithelial cells. To improve DGC diagnostics, we propose using Deep Learning (DL) to aid with the detection of relevant cell types in H&E-stained slides of gastric biopsies and resections.

Training fully-supervised DL methods requires exhaustive cell annotations, which is expensive and places a large burden on expert pathologists. To address this issue, we propose using *interactive object detection*, a technique that involves both humans and AI in the annotation process. Specifically, we build on the recent work of (Lee et al., 2022) who proposed a generic interactive multi-class object detection method called *C3Det*. The resulting workflow consists of an annotator making a few initial annotations (clicks) in the image, followed by the C3Det model making object proposals based on these user input clicks. Subsequently, the annotator can confirm or deny the proposals, and make additional annotations. This process can be repeated iteratively until annotation is completed.

To achieve this workflow, a generic convolutional neural network architecture is developed that takes as input an image and additionally a set of *user inputs*. In our work, we implemented this framework for the detection of relevant cell types in DGC. To this end, we trained C3Det on a dataset consisting of H&E whole-slide images (WSIs) of DGC patients, evaluated the resulting model and showed that it substantially decreases the amount of user actions required to achieve exhaustive annotation.

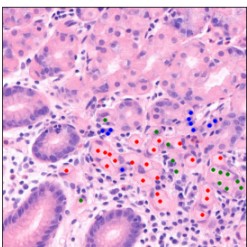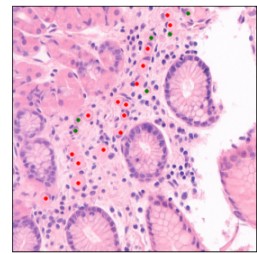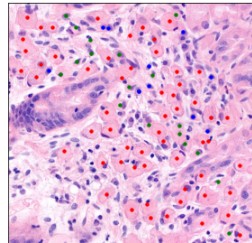

Figure 1: Example patches from the training set, with reference standard annotations in the color scheme src: red, src_s: green, pc: blue.

## 2. Methodology

**Data.** We collected a set of 108 H&E-stained WSIs of 9 different patients which were collected between 2020 and 2022 at Radboud university medical center, the Netherlands. Slides were scanned at 0.25 micron per pixel spacing. The 9 patients are split across train (4), validation (3) and test (2) sets. To develop the model we used a fully manual annotation approach where four annotators exhaustively annotated three cell types in the 108 WSIs: classical SRCs (src), small SRCs (src_s), and poorly differentiated cells (pc). Annotations consist of point annotations placed near the center of a cell. Training the C3Det model requires bounding box annotations, however. To convert the point annotations to bounding boxes, we developed and applied a conversion pipeline using NuClick (Alemi Koohbanani et al., 2020). Next, regions of interest were extracted from the WSIs, which were further divided into tiles of 1200×1200 pixels. The resulting dataset contains 629, 21, and 120 patches in the train, validation, and test sets, respectively. Three example patches are shown in Figure 1.

**Model development and validation.** The trained model uses a Faster-RCNN architecture with a ResNet-50 backbone. We trained for 60 epochs with a starting learning rate of 0.0001 and a linear decay schedule, decreasing the learning rate by a factor of 10 at epochs 15, 30, and 45. To validate the model, we performed inference on the held-out test set after simulating multiple values of user clicks (*noc*) by sampling annotations from the manual reference standard, and calculated the F1 scores for each value of *noc*. Furthermore, we investigated the advantage of using the model in an interactive annotation workflow by comparing the required user actions to reach full annotation in the manual and interactive approaches. The interactive approach consists of a single iteration of the interactive workflow: an annotator makes *noc* initial clicks, the model makes proposals based on these inputs, and the annotator removes the false positives and adds annotations for the false negatives. In this approach, we calculated the number of required user actions as the sum of initial clicks, false positives, and false negatives, and we report the difference between this metric and the number of ground truths for all *noc*. This difference represents the reduction in user actions to reach full annotation.

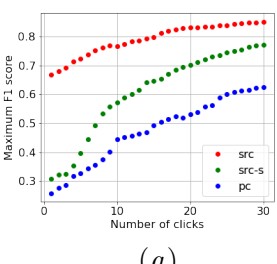
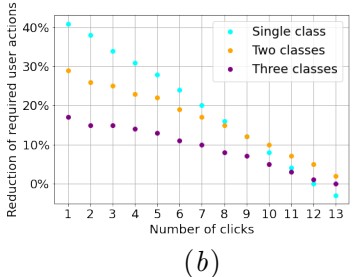
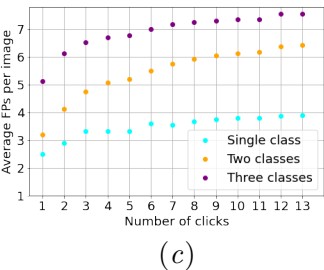

$(a)$ $(b)$ $(c)$

Figure 2: (a): Maximum F1 score as a function of number of user input clicks.
(b): Reduction of required user actions to achieve full annotation, comparing the interactive to the manual approach. Results are presented for up to 14 initial clicks, as a further increase in the number of clicks leads to an increase in required user actions.
(c): Average number of false positives per image.

## 3. Results

We measured the performance of the model in terms of the F1 score for each class. The maximum F1 score is shown in Figure $2(a)$ as a function of the number of user input clicks. As expected, F1 scores increase with more user input. However, we observe a substantial difference in performance for poorly differentiated cell detection compared to the other classes, which can be attributed to the unbalanced dataset and the difficulty of detecting this class. The advantage of using our interactive annotation approach is demonstrated in Figure $2(b)$, where we see a substantial reduction in the number of required user actions to achieve full annotation. The maximum reduction is 17% for the multi-class case, 29% if we combine the src and src_s classes, and 41% if we discard the pc class. Interestingly, the largest reduction is achieved for a low number of initial user inputs. This is because the average number of false positives does not decrease as we increase the number of user inputs, as shown in Figure $2(c)$. This behavior may also limit the overall model performance as measured by the F1 score.

## 4. Conclusion

In this work we present a novel approach to improve the annotation workflow for the detection of signet ring cells and poorly differentiated cells in slides of patients with DGC. To our knowledge, this is the first model developed for the interactive detection of these cell types. The model achieves promising performance and is shown to be able to substantially reduce the effort required to annotate signet ring cell carcinomas.

The reduction in user actions shown in Figure $2(b)$ assumes a single iteration of model inference. In an ideal setting, however, multiple iterations of the model would run, with the annotator only removing false positives (FPs) after each iteration and feeding the new predictions as user inputs to the next iteration. Additionally, a filter could be implemented to remove recurring FPs after the annotator marks them as such. This approach could alleviate the issue of increasing FPs with an increasing number of user inputs illustrated in Figure $2(c)$, and result in further reduction of annotation costs.

## Acknowledgments

This research was supported by an unrestricted grant of Stichting Hanarth Fonds, The Netherlands. In addition, this project has received funding from the European Union's Horizon 2020 research and innovation programme under grant agreement No 825292 (ExaMode, htttp://www.examode.eu/).

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
