# OpenReview forum: "Interactive Cell Detection in H&E-stained slides of Diffuse Gastric Cancer"
_MIDL.io/2023/Short_Paper_Track — MIDL 2023 Short paper track Poster_

### Official Review · Reviewer_LLj2 · 2023-04-23
**This paper presents an interactive detection model using deep learning is proposed for improved cell annotation in diffuse gastric cancer histology, reducing user actions needed for image annotation by 17%.**

**Rating:** 7
**Confidence:** 5

**Review:**

This paper presents a convolutional neural network (CNN) architecture for the interactive detection of three types of cells in diffuse gastric cancer (DGC) histology. The proposed model is built upon the recent work of C3Det and aims to improve the cell annotation workflow by reducing the number of user actions required for complete image annotation by 17%. The results show promising potential for enhancing DGC diagnostics and reducing the workload of expert pathologists.

**Pros:**
- The paper is written clearly.
- The paper presents an interesting approach based on the existing method (C3Det) to reduce the interaction time for annotating histology images.
- The model was tested on a dataset of 108 H&E-stained whole-slide images (WSIs) from 9 different patients, demonstrating its applicability.
- A 17% reduction in annotation time was reported for the test data, indicating the effectiveness of the proposed approach.

**Cons:**

- The main limitation of the study is the small size of the testing set, and the authors should consider testing and evaluating the method on larger cohorts of patients to validate its robustness.
- It would be beneficial to evaluate the proposed approach on images from different institutions and with varying image resolutions to assess its generalizability.
- As mentioned by the authors, the model runs multiple iterations to remove false positives, which requires further analysis to understand its impact on performance.
- An ablation study should be conducted to train the model with diverse CNN backbones, which could potentially improve the detection results and reduce false positives.

---

### Official Review · Reviewer_CJRZ · 2023-04-25
**The method used in the paper is creative and practical, but the innovation is not outstanding, and there are some ambiguities and flaws in the method description.**

**Rating:** 4
**Confidence:** 4

**Review:**

The main advantages of the paper are: 1) Innovative application of interactive object detection model for DGC cell detection; 2) Through a series of experiments, it has been demonstrated that the proposed interactive annotation method can significantly reduce the annotation workload and has certain practical value. The main disadvantages of the paper are: 1) Insufficient introduction of the method section, and the specific network model was not introduced, which may make it difficult for readers to understand; 2) The innovation is not specifically demonstrated, and there is little improvement compared to previous works that were used as references; 3) Insufficient details were provided in the description of Figure 2(b) and 2(c), and the specific meaning of the different colored dots was not clear.